

# Mitochondrial DNA and Y chromosome reveal the genetic structure of the native Polish Konik horse population

Adrianna Dominika Musiał[1], Lara Radović[2,3], Monika Stefaniuk-Szmukier[1], Agnieszka Bieniek[1], Barbara Wallner[2] and Katarzyna Ropka-Molik[1]

[1] Department of Animal Molecular Biology, National Research Institute of Animal Production, Balice, Poland
[2] Institute of Animal Breeding and Genetics, University of Veterinary Medicine Vienna, Vienna, Austria
[3] Vienna Graduate School of Population Genetics, University of Veterinary Medicine Vienna, Vienna, Austria

Corresponding author
Adrianna Dominika Musiał,
adrianna.musial@iz.edu.pl

## ABSTRACT

Polish Konik remains one of the most important horse breeds in Poland. The primitive, native horses with a stocky body and mouse-like coat color are protected by a conservation program, while their Polish population consists of about 3,480 individuals, representing 16 dam and six sire lines. To define the population's genetic structure, mitochondrial DNA and Y chromosome sequence variables were identified. The mtDNA whole hypervariable region analysis was carried out using the Sanger sequencing method on 233 Polish Koniks belonging to all dam lines, while the Y chromosome analysis was performed with the competitive allele-specific PCR genotyping method on 36 horses belonging to all sire lines. The analysis of the mtDNA hypervariable region detected 47 SNPs, which assigned all tested horses to 43 haplotypes. Most dam lines presented more than one haplotype; however, five dam lines were represented by only one haplotype. The haplotypes were classified into six (A, B, E, J, G, R) recognized mtDNA haplogroups, with most horses belonging to haplogroup A, common among Asian horse populations. Y chromosome analysis allocated Polish Koniks in the Crown group, condensing all modern horse breeds, and divided them into three haplotypes clustering with coldblood breeds (28 horses), warmblood breeds (two horses), and Duelmener Pony (six horses). The clustering of all Wicek sire line stallions with Duelmener horses may suggest a historical relationship between the breeds. Additionally, both mtDNA and Y chromosome sequence variability results indicate crossbreeding before the studbooks closure or irregularities in the pedigrees occurred before the DNA testing introduction.

## INTRODUCTION

Polish Konik (Polish Primitive Horse; Fig. 1) is one of the most important native horse breeds of Poland. The Polish Konik horse, like many other primitive breeds, presents traits such as good health and high fertility, resilience, and adaptability to life in forest conditions (*Hendricks, 2007*; *Doboszewski et al., 2017*). Its body is strong and stocky, with a primitive mouse-grey coat color and characteristic black stripe along the back (*Janczarek, Pluta & Paszkowska, 2017*). The Polish Koniks are good working horses and are especially valued in gardening farms (*Hendricks, 2007*), where the beneficial effects of grazing have been proven for plants and rare bird populations (*Doboszewski et al., 2017*). They also take a part in nature conservation being used for rewilding processes and habitat protection (*Reke, Zarina & Vinogradovs, 2019*). On the other hand, due to its gentleness and docility, this breed is also used for riding (*Hendricks, 2007*). Thanks to its advantages, Polish Konik studs can be found not only in Poland but also abroad, under stable and reserve management or used as recreational mounts in countries such as the Netherlands, Germany, Latvia, Belgium, and the UK (*Gorecka-Bruzda et al., 2020*).

Polish Konik horses are bred in traditional studs and semi-feral/free-roaming groups like in the forest sanctuary of Popielno Research Station (Poland) and are considered a valuable genomic resource (*Gorecka-Bruzda et al., 2020*; *Wolc & Balińska, 2010*). Although the population has increased in recent years, it is still limited. Due to their low census population size, genetic resource conservation programs are established to maintain population size, genetic diversity, and preserve this endangered breed (*Jaworski & Tomczyk-Wrona, 2019*; *Pasicka, 2013*).

Despite the importance of this native breed, the origin of the Polish Konik has not been clearly explained so far. The most common theory assumes that Polish Koniks are the direct descendent of the European wild horse—the Tarpan, a species that inhabited the forests of Europe and became extinct in the 19th century (*Pasicka, 2013*). However, in recent years, the direct descent of the Polish Konik from the Tarpan has been highly debated. *Lovasz, Fages & Amrhein (2021)* suggest that it is a manmade myth that hinders effective breed conservation management. Recent advances in the field of ancient DNA analysis showed that the Tarpan, referred to as the last wild horse, was also a representative of a cluster of domestic horses that spread from 2,200 BC onwards to the second millennium BC, and were a source of modern horses. The results shattered the common hypothesis of Tarpans as the wild ancestors of modern horses or hybrids with Przewalski horses (*Librado et al., 2021*), making the uncovering of Polish Konik origin an important and difficult challenge to solve.

Although a century has passed since the Polish Konik began to be the object of interest of researchers, there is still no clear information about the origin of the breed. The first research conducted on the Polish Koniks was carried out in 1914 and suggested the presence of primitive horses, resembling Tarpans, near the city of Biłgoraj (Lublin Voivodeship, Poland). In the 1920s, the name "Polish Konik" was introduced into literature by Prof. Tadeusz Vetulani, who played an outstanding role in starting the stud

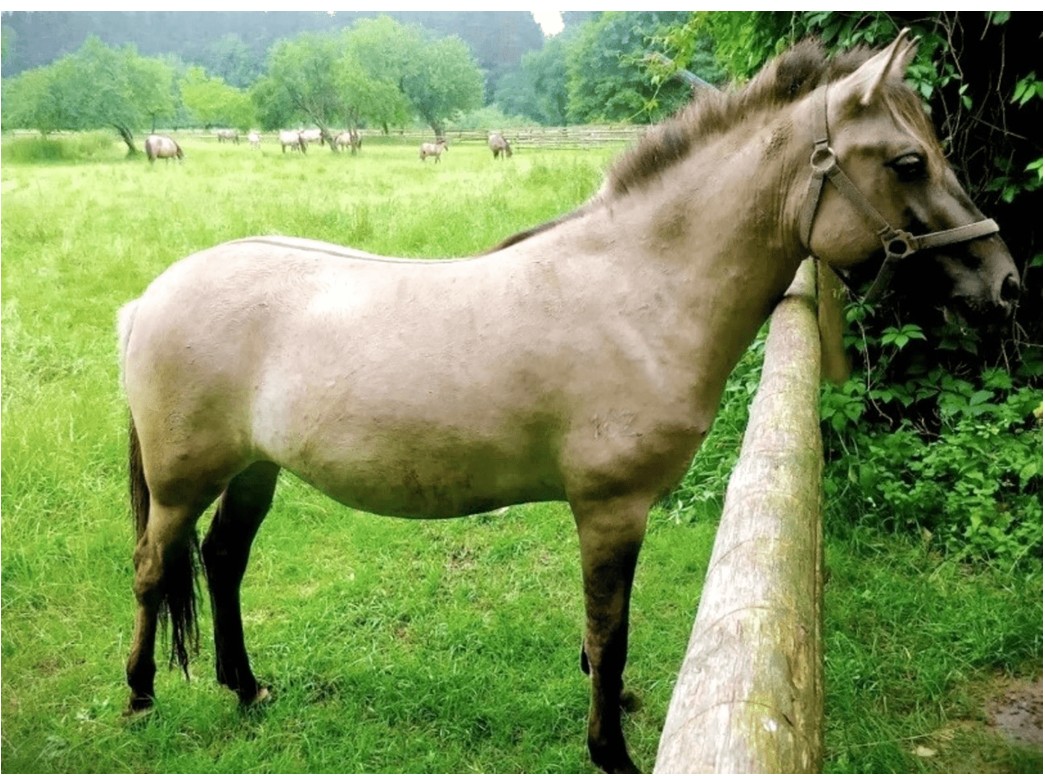

**Figure 1 Polish Konik in Roztoczański National Park, Poland (photo credit: Adrianna Musiał).**

breeding of horses. The history of breeding dates back to 1923, when Polish Koniks were placed in the Janów Podlaski stud, and the following years saw the opening of several new studs. Unfortunately, the progress in breeding was interrupted by World War II, when most of the Polish Koniks were taken to Germany or were lost (*Tomczyk-Wrona, 2022*). After the end of the war, plenty of efforts were made to restore the population of Polish Koniks and resume their breeding; however, most of the horses did not have documented origins. The supervision of the Polish Koniks breeding led to the publication of the first "Studbook of Polish Koniks" in 1962, and in 1984 the studbooks became closed, which eliminated the possibility of crossing with individuals of other horse breeds, and breeding is conducted in a pure breed (*Mackowski et al., 2015*; *Tomczyk-Wrona, 2022*).

The current population of the Polish Konik registered in the studbook (data for 2022) consists of ~3,480 individuals: ~1,760 mares, 183 stallions, and ~1,540 foals. Over the last 15 years, the number of stallions has not changed significantly; a slight increase in their numbers was noted at the beginning of the 21st century. In the case of Polish Konik mares, the number of horses has doubled since 2010 and has almost quadrupled since 2005 due to the breeding approach and subsidies granted mainly for breeding mares (*Polish Horse Breeders Association, 2022*). All Polish Koniks come from 35 female lines, of which 19 became extinct and only 16 are currently active: Liliputka I (1920), Karolka (1933), Zaza (1933), Urszulka (1934), Tarpanka I (1937), Traszka (unknown), Tunguska (1949),

Tygryska (1928), Popielica (1937), Wola (1943), Białka (1944), Ponętna (1946), Misia II (1948), Dzina I (unknown), Bona (1954), and Geneza (1965); (*Jaworski, 1997*). The first seven lines showed a clear progression in the last years, while the remaining lines show only slight progression or stagnation. Horses belonging to the three most numerous dam lines (Traszka, Tarpanka I, and Zaza) make up more than half of all the individuals. Due to the persistent state of stagnation, the lines originating from the mares Bona, Ponętna, Misia II, Geneza, and Białka are threatened with extinction. The Polish Konik males are distinguished into six lines, which were founded by the stallions: Chochlik (1940), Goraj (1935), Glejt I (1944), Wicek (<1930), Liliput (1918), and Myszak (1937), of which the Liliput and Glejt I lines are the least numerous (*Tomczyk-Wrona, 2022*).

Recently, mitochondrial DNA and Y chromosome diversity studies using Single Nucleotide Polymorphism (SNP) have started to play an increasing role in population genetics including horse genetics. The genetic information is transmitted as a single haplotype (HT) block, which mirrors dam and sire line history. Thus, we can easily detect pedigree incongruences since all individuals from a certain line (sire or dam) should have the same haplotype (*Hutchison et al., 1974*; *Liu, 2010*).

The polymorphisms present in the mitochondrial DNA D-loop sequence made it possible to distinguish the dam lines and identify the genetic diversity, as well as maternal ancestry and relationships, in many different horse breeds, like Arabian horses (*Khanshour & Cothran, 2013*), Thoroughbreds (*Yoon et al., 2018*), Cleveland Bay horses (*Dell et al., 2020*), Tibetan horses (*Yang et al., 2018*), Holstein horses (*Engel et al., 2021*), Polish Draft horses (*Myćka et al., 2022*), or Hucul horses (*Czerneková, Kott & Majzlík, 2013*). The part of the mitochondrial DNA used in most diversity research is hypervariable region 1 (HVR1), localized in the D-loop. Since the early 2000s, single individuals of Polish Konik have started to be included in multi-breed studies of mtDNA variation (*Cothran, Juras & Macijauskiene, 2005*; *Kusza et al., 2013*; *Cieślak et al., 2017*). However, a complete analysis of all present Polish Konik dam lines has not been performed so far.

Regarding paternal ancestry, the researchers focus on the Y chromosome male-specific region (MSY), which is inherited without recombination from fathers to sons (*Jobling & Tyler-Smith, 2017*) and allows tracing the sire lines (*Felkel et al., 2019*; *Radovic et al., 2021*). Previous efforts established the possibility of exploiting the variation present in the MSY region to identify the diversity of sire lines as well as illustrate male lineages' demographic history (*e.g.*, *Remer et al., 2022*). Currently, MSY variation (>3,000 variants) groups modern domesticated stallions tested so far into one 1,500-year-old haplogroup–the Crown (daC). Only a handful of remote horse breeds carry haplotypes beyond the Crown (*Bozlak et al., 2023*; *Felkel et al., 2018*), classified as Non Crown.

Here, we investigate the current genetic diversity of Polish Konik maternal and paternal lines using uniparental markers. The aim of this research was the investigation of the Polish Konik population structure in terms of dam and sire lines, based on the analysis of the mtDNA whole D-loop sequence as well as the analysis of selected Y chromosomal markers.

## MATERIALS AND METHODS

### Founder lines sampling

The variability of mitochondrial DNA D-loop sequence was examined in 233 Polish Koniks belonging to all 16 dam founding lines: Białka (nine individuals), Dzina I (14), Geneza (six), Karolka (17), Liliputka I (19), Misia II (six), Ponętna (10), Popielica (11), Tarpanka I (20), Traszka (44), Tunguska (13), Tygryska (seven), Urszulka (22), Wola (11), Zaza (21), and Bona (three). The number of analysed horses constituted approximately 7.3% of the total active Polish Konik population, and the share of individual dam lines included in the research was also proportional to the total number of horses belonging to each dam line. MSY variability was analysed in 36 stallions selected from the individuals above and belonged to all six sire founding lines: Goraj (six individuals), Liliput (six), Wicek (six), Glejt I (six), Chochlik (six), and Myszak (six). The lines were assigned to the horses based on pedigree data, and all analysed animals were available in the pedigree database of the Polish Horse Breeders Association. The material used in this study were hair follicles and blood samples that came from a genetic material bank of the National Research Institute of Animal Production, Poland.

### DNA isolation

The DNA extraction was made using the Sherlock AX kit (A&A Biotechnology, Gdańsk, Poland) according to the manufacturer's protocol, and DNA quality was determined by checking concentration and purity on the NanoDrop 2000 spectrophotometer (Thermo Fisher Scientific, Waltham, MA, USA). DNA samples with concentration >40 ng/μl and purity (A260/A280) between 1.8–2.0 were qualified for the next steps and stored at −20 °C.

### Mitochondrial DNA analysis

For the amplification of the mitochondrial DNA D-loop hypervariable region 1 and hypervariable region 2, three pairs of primers were designed based on the GenBank *Equus caballus* reference sequence: NC_001640 (Table 1). PCR products covered a total region of 1,062 bp. Amplification was performed on all 233 individuals using the Phanta Ready Mix (Vazyme Biotech, Nanjing, China) according to the instructions. To check the specificity of the obtained products, the separation on the 3% agarose gel with the ethidium bromide addition, along with a DNA length marker–Marker1 100–1,000 bp (A&A Biotechnology, Gdańsk, Poland), was performed (120 V, 30 min.). PCR products were cleaned from primers and free nucleotides with the enzymatic method using the EPPiC Fast reagent (A&A Biotechnology, Gdańsk, Poland) and used as a template for sequencing by the Sanger method. The PCR for the sequencing reaction was performed on 233 individuals for each amplicon (699 samples in total) with the BigDye Terminator v3.1 Cycle Sequencing Kit (Thermo Fisher Scientific, Waltham, MA, USA) according to the instructions, and the products were repurified with the BigDye XTerminator Purification Kit (Thermo Fisher Scientific, Waltham, MA, USA) according to the protocol. The capillary electrophoresis was performed on 3500xL Genetic Analyzer (Thermo Fisher Scientific, Applied Biosystems, Foster City, CA, USA) using POP-7™ Polymer for 3500/3500xL (Thermo Fisher Scientific, Applied Biosystems, Foster City, CA, USA). The results

**Table 1 Primer sequences and length of the amplicons.**

| Primer | Sequences (5′→ 3′) | Length (bp) |
|---|---|---|
| amp1 | F: AACGTTTCCTCCCAAGGACT<br>R: GTAGTTGGGAGGGTTGCTGA | 397 |
| amp2 | F: ACCCCATCCAAGTCAAATCA<br>R: CAGGTGCACTTGTTTCCTATG | 462 |
| amp3 | F: ACCTACCCGCGCAGTAAGCAA<br>R: ACGGGGGAAGAAGGGTTGACA | 306 |

in the form of chromatograms were analysed using BLAST (*Altschul et al., 1990*), FinchTV 1.3.0 (Geospiza, Inc., Seattle, WA, USA), DnaSP 5.10.01 (*Librado & Rozas, 2009*), and Variant Analysis application (Thermo Fisher Scientific Cloud, Waltham, MA, USA) and compared with a GenBank reference sequence NC_001640.

The SNPs made it possible to identify individual haplotypes and all the sequences were subjected to phylogenetic analysis. The phylogenetic tree illustrating the variability of analysed mitochondrial DNA fragments was constructed using Mega 11.0.13 software (*Tamura, Stecher & Kumar, 2021*) and iTOL 6.8.1 (*Letunic & Bork, 2021*) with the neighbor-joining method (*Saitou & Nei, 1987*), including 1,000 bootstrap replications. To visualize the connections between all mtDNA HTs of Polish Koniks, the median-joining network (*Bandelt, Forster & Röhl, 1999*) was constructed in the PopART phylogenetic software (https://popart.maths.otago.ac.nz/, accessed September 2023). Haplotypes found within the studied Polish Konik population were compared with sequences stored in the GenBank database using BLAST (*Altschul et al., 1990*), and the similarities between Polish Konik population HTs with other breeds have been described. The sequences representing haplogroups published by *Achilli et al. (2012)* were found among the BLAST results. The sequence with the highest similarity percentage was selected, and its haplogroup was checked to categorize the identified Polish Konik haplotypes into the haplogroups.

## Y chromosome analysis

Y chromosomal haplotypes were determined for 36 Polish Koniks, representing all six founding sire lines. For haplotyping, a downscaled haplotype structure of the most recent horse Y phylogeny was constructed (*Bozlak et al., 2023*) based on 114 selected variants (92 in the Crown and 22 Non-Crown, see Table S1) that determine 115 HTs. In a hierarchical manner (as described by *Remer et al. (2022)*), allelic states of variants in the downscaled haplotype structure were determined *via* the competitive allele-specific PCR (KASP™, www.lgcgroup.com) genotyping method under the standard KASP™ genotyping protocol (lgcgroup.com). The analysis was performed on a CFX96 Touch® BioRad Real-Time PCR machine. In addition to Polish Konik samples, each reaction included two positive controls with known allelic states and two negative controls, including female DNA and water. In two individuals, the tetranucleotide microsatellite fBVB (GATA14/GATA15) was screened following *Felkel et al. (2019)* and *Remer et al. (2022)*.
Raw data were analysed with Bio-Rad CFX Manager 3.1® software (BioRad). The allelic states of all tested variants were catenated, while the allelic states of variants that were not tested were imputed according to the published HT structure (*Bozlak et al., 2023*, see Table S2). For visualization, a median-joining haplotype network was constructed with the program Network 10.2 (*Bandelt, Forster & Röhl, 1999*), and the output was redrawn as a haplotype frequency plot with Canva Pro (Canva, https://www.canva.com/pro/, accessed August 2023).

# RESULTS

## Mitochondrial DNA variability

The whole mitochondrial DNA hypervariable region 1 and hypervariable region 2 sequences were determined in 233 Polish Konik horses using Sanger sequencing of three PCR products (Table S3). The alignment of three Sanger sequencing products revealed the entire mitochondrial DNA hypervariable region for each horse, allowing comparison with a publicly available reference sequence (NC_001640) and between all samples. Cutting the obtained sequences from the location of the first identified SNP (15,542 bp) to the last one (16,611 bp) led to the receiving of 1,070 bp fragments, whose analysis detected 47 variable sites. The analysis in DnaSP indicated a nucleotide diversity result of 0.010, and the average number of nucleotide differences of 11.104. The 43 mtDNA HTs were defined in Polish Konik samples and reported to the GenBank database, where received accession numbers OR827103–OR827145 (Table 2). The haplotype diversity amounted to 0.96.

Among the considered dam lines, 11 of 16 lines presented more than one HT (Table 2). The exceptions were Białka, Bona, Dzina I, Popielica, and Tunguska maternal lines, in which all examined individuals presented the same, unique haplotype. On the other hand, in both Traszka and Zaza lines, there were six different HTs. Some of the haplotypes were recognized in two different lines: HT9 in Karolka and Ponętna lines, HT33 in Tygryska and Zaza, HT6 in Karolka and Liliputka I, HT20 in Tarpanka I and Zaza, and HT19 in Tarpanka I and Popielica. Compared to the reference sequence, the most nucleotide differences were found in HT18 (Ponętna) and HT39 (Wola) HTs–22 SNPs. Only one nucleotide difference from the reference sequence was found in HT22 (Tarpanka) and HT30 (Tunguska) HTs. Out of 47 detected SNP positions, there was one insertion position (g.16557_16558insC), one transversion position (g.16543 T > A), one position with both transversion and transition (g.15776 T > C, T > G; Fig. 2), and transitions in the other 44 positions.

The obtained Polish Konik HTs were compared with sequences available in GenBank showing that none of them presents 100% coverage. The similarity to Polish Konik HTs of at least 99% has been presented by a whole range of different horse breeds, such as Arabian horses (KU575096.1), Polish Draft horses (ON052712.1), Akhal Teke horses (JN398385), Thoroughbred horses (KC202956.1), Selle Francais horses (OR909772.1), East Asian horses (JQ340108.1), Chinese Native horses (JQ710930.1), and also Przewalski horses (ON393916.1).

To construct the neighbor-joining tree (Fig. 3), supported by bootstrap percentages (BP) computed with 1,000 replicates, we used 233 sequences representing all 16 Polish Konik

**Table 2 Mitochondrial DNA variants located between positions 15,542 and 16,611 detected in all Polish Konik haplotype sequences compared to the reference sequence NC_001640.** The number of horses belonging to each haplotype is included. The HTs presented by more than one dam line are underlined. All obtained HTs were deposited in the GenBank database and received accession numbers OR827103–OR827145.

| Dam line/HT | | GenBank acc. no. | HG* | No. of horses | 15542 | 15585 | 15597 | 15598 | 15601 | 15602 | 15604 | 15615 | 15616 | 15617 | 15635 | 15650 | 15659 | 15666 | 15667 | 15672 | 15703 | 15762 | 15766 | 15769 | 15770 | 15771 | 15775 | 15776 | 15777 | 15806 | 15807 | 15810 | 15811 | 15826 | 15827 | 15868 | 15870 | 15871 | 15956 | 15974 | 15995 | 15996 | 16007 | 16022 | 16540 | 16543 | 16546 | 16551 | 16557_16558insC | 16559 | 16611 |
|---|---|---|---|---|---|---|---|---|---|---|---|---|---|---|---|---|---|---|---|---|---|---|---|---|---|---|---|---|---|---|---|---|---|---|---|---|---|---|---|---|---|---|---|---|---|---|---|---|---|
| | | Ref: NC_001640 | | | C | G | A | T | T | C | G | A | A | T | C | A | T | G | A | G | T | G | C | T | C | C | C | T | A | C | C | A | C | A | A | T | C | C | A | C | A | T | T | T | C | T | T | G | - | C | A |
| Białka | HT1 | OR827103 | E | 9 | T | . | G | . | . | T | . | . | . | . | . | G | . | A | . | . | . | . | . | . | . | . | . | . | . | . | . | . | . | . | . | T | . | G | T | . | . | . | . | . | . | . | . | . | . | . | . |
| Bona | HT2 | OR827104 | R | 3 | . | . | . | C | T | . | G | G | . | . | . | C | . | . | . | C | . | . | . | T | . | T | C | . | T | . | . | . | . | G | . | . | . | G | T | . | C | . | . | T | A | . | . | . | . | . | . |
| Dzina I | HT3 | OR827105 | A | 14 | . | A | . | . | C | T | . | . | . | . | . | . | . | . | . | . | . | . | . | . | . | . | . | . | . | . | . | . | . | . | . | . | . | G | . | . | . | . | . | . | A | C | . | . | . | T | . |
| Geneza | HT4 | OR827106 | E | 5 | T | . | G | . | . | T | . | . | . | . | . | G | . | A | . | . | . | . | . | . | . | . | . | . | T | . | . | . | . | . | . | T | . | G | T | . | . | . | . | . | . | . | . | . | . | . | . |
| Geneza | HT5 | OR827107 | A/B | 1 | T | . | G | . | . | T | . | . | . | . | . | G | . | A | . | . | . | . | . | . | . | . | . | . | . | . | . | . | . | . | . | . | . | G | . | . | . | . | . | . | . | . | . | . | . | . | . |
| Karolka | HT6 | OR827108 | A | 10 | . | . | . | . | . | . | . | . | . | . | . | . | . | . | . | . | . | . | . | . | . | . | . | . | . | . | . | . | . | . | . | . | . | G | . | . | . | . | . | . | . | . | . | . | . | C | . |
| Karolka | HT7 | OR827109 | A | 2 | . | A | . | . | C | T | . | . | . | . | . | . | . | . | . | . | . | . | A | T | C | T | . | T | G | . | . | T | . | T | . | G | . | T | . | G | . | G | . | C | C | . | A | C | . | . | T |
| Karolka | HT8 | OR827110 | G | 2 | T | . | G | . | . | T | . | . | . | T | G | . | A | . | A | C | . | . | . | . | . | . | . | . | . | . | . | . | . | . | . | . | . | G | . | . | . | . | . | . | . | . | . | . | . | . | . |
| Karolka | HT9 | OR827111 | G | 2 | T | . | G | . | . | T | . | . | . | T | G | . | A | . | A | C | . | . | . | . | . | . | . | . | . | . | . | . | . | . | . | . | T | T | . | . | . | . | . | . | . | . | . | . | . | . | . |
| Karolka | HT10 | OR827112 | A | 1 | . | A | . | . | . | T | . | . | . | . | . | . | . | . | . | . | . | C | . | . | . | . | . | . | . | . | . | . | . | . | . | . | . | G | . | . | . | . | . | . | . | A | . | . | . | . | . |
| Liliputka I | HT11 | OR827113 | B | 13 | . | A | . | . | . | . | . | . | . | . | . | . | G | . | A | . | . | . | . | . | . | . | . | . | . | . | . | . | G | . | G | . | . | . | . | . | . | . | . | . | . | . | . | . | . | . | . |
| Liliputka I | HT6 | OR827108 | A | 3 | . | . | . | . | . | . | . | . | . | . | . | . | . | . | . | . | . | . | . | . | . | . | . | . | . | . | . | . | . | . | . | . | . | G | . | . | . | . | . | . | . | . | . | . | . | C | . |
| Liliputka I | HT12 | OR827114 | A/B | 3 | . | . | . | . | . | . | . | . | C | . | G | . | A | . | . | . | . | . | . | . | . | . | . | . | . | . | . | . | . | . | . | . | . | G | . | . | . | . | . | . | . | . | . | . | . | . | . |
| Misia II | HT13 | OR827115 | A | 3 | . | A | . | . | . | T | . | . | . | . | . | . | . | . | . | . | . | . | . | T | . | . | . | . | . | . | T | . | . | . | . | . | G | . | T | T | . | . | . | . | . | . | . | . | . | . | . |
| Misia II | HT14 | OR827116 | R | 2 | . | . | . | C | T | . | G | G | . | . | . | C | . | . | . | C | . | . | . | T | . | T | . | T | . | . | . | . | . | G | . | . | . | T | G | T | . | C | . | . | T | A | . | . | . | . | . |
| Misia II | HT15 | OR827117 | A | 1 | . | . | . | C | T | . | G | G | . | . | . | C | . | . | . | . | . | . | . | . | . | . | . | . | . | . | . | . | . | G | . | . | T | . | . | . | . | . | C | T | A | . | . | . | . | . | . |
| Ponętna | HT16 | OR827118 | A | 4 | . | A | . | . | . | T | A | . | . | . | . | . | . | . | . | C | . | . | . | . | . | . | . | . | . | . | . | . | . | G | . | . | . | . | . | . | . | . | . | A | . | . | C | . | . | . | . |
| Ponętna | HT17 | OR827119 | A | 3 | . | A | . | . | . | T | A | . | . | . | . | . | . | . | . | C | . | . | . | . | . | . | . | . | . | . | . | . | . | . | T | . | . | . | . | . | . | . | . | A | . | . | C | . | . | . | . |
| Ponętna | HT18 | OR827120 | A | 2 | . | A | . | . | . | T | A | . | . | . | . | . | . | . | . | C | A | T | C | T | . | T | G | . | . | T | . | T | . | G | C | T | . | G | . | G | . | C | C | . | A | . | . | C | . | . | G |
| Ponętna | HT9 | OR827111 | G | 1 | T | . | G | . | . | T | . | . | . | T | G | . | A | . | A | C | . | . | . | . | . | . | . | . | . | . | . | . | . | . | . | T | . | . | . | . | . | . | . | . | . | . | . | . | . | . | . |
| Popielica | HT19 | OR827121 | J | 11 | . | A | . | . | . | T | . | . | . | . | . | . | . | . | . | . | . | . | . | T | . | . | . | . | . | T | . | . | . | G | . | T | T | . | . | . | . | . | . | . | . | . | . | . | . | . | . |
| Tarpanka I | HT20 | OR827122 | A | 9 | . | A | . | . | . | T | A | . | . | . | . | . | . | . | . | C | . | . | . | . | . | . | . | . | . | . | . | . | . | G | . | . | . | . | . | . | . | . | . | A | . | . | . | . | . | . | . |
| Tarpanka I | HT21 | OR827123 | A | 6 | . | A | . | . | C | T | . | . | . | . | . | . | . | . | . | . | . | . | . | . | . | . | . | . | . | . | . | . | . | G | . | . | . | . | . | . | . | . | . | A | C | . | . | T | . | . | . |
| Tarpanka I | HT22 | OR827124 | A | 3 | . | A | . | . | . | T | . | . | . | . | . | . | . | . | . | . | . | . | . | . | . | . | . | . | . | . | . | . | . | G | . | . | . | . | . | . | . | . | . | . | . | . | . | . | . | . | . |
| Tarpanka I | HT19 | OR827121 | J | 1 | . | A | . | . | . | T | . | . | . | . | . | . | . | . | . | . | . | . | . | T | . | . | . | . | . | T | . | . | . | G | . | T | T | . | . | . | . | . | . | . | . | . | . | . | . | . | . |
| Tarpanka I | HT23 | OR827125 | A/B | 1 | T | . | G | . | . | T | . | . | . | . | . | G | . | A | . | . | . | . | . | . | . | . | . | . | . | . | . | . | . | . | . | . | . | G | . | . | . | . | . | . | A | . | . | . | . | . | . |

Musiał et al. (2024), PeerJ, DOI 10.7717/peerj.17549

| Dam line/HT | | GenBank acc. no. | HG* | No. of horses | 15542 | 15585 | 15597 | 15598 | 15601 | 15602 | 15604 | 15615 | 15616 | 15617 | 15635 | 15650 | 15659 | 15666 | 15667 | 15672 | 15703 | 15762 | 15766 | 15769 | 15770 | 15771 | 15775 | 15776 | 15777 | 15806 | 15807 | 15810 | 15811 | 15826 | 15827 | 15868 | 15870 | 15871 | 15956 | 15974 | 15995 | 15996 | 16007 | 16022 | 16540 | 16543 | 16546 | 16551 | 16557_16558insC | 16559 | 16611 |
|---|---|---|---|---|---|---|---|---|---|---|---|---|---|---|---|---|---|---|---|---|---|---|---|---|---|---|---|---|---|---|---|---|---|---|---|---|---|---|---|---|---|---|---|---|---|---|---|---|---|---|---|
| | | Ref: NC_001640 | | | C | G | A | T | T | C | G | A | A | T | C | A | T | G | A | G | T | G | C | T | C | C | C | T | A | C | C | A | C | A | A | T | C | C | A | C | A | T | T | T | C | T | T | G | - | C | A |
| Traszka | HT24 | OR827126 | A | 26 | . | . | . | . | . | T | . | . | . | C | . | . | C | . | . | . | . | . | . | . | . | . | . | . | . | . | . | . | . | . | . | . | . | . | G | . | . | . | . | . | . | A | C | . | . | T | . |
| Traszka | HT25 | OR827127 | A | 9 | . | . | . | . | . | T | . | . | . | C | . | . | C | . | . | . | . | . | . | . | . | . | . | . | . | . | . | . | . | . | . | . | . | . | G | . | . | . | . | . | . | A | C | . | . | T | . |
| Traszka | HT26 | OR827128 | A | 3 | . | . | . | . | . | T | . | . | . | C | . | . | C | . | . | . | C | A | T | C | T | . | T | G | . | . | T | . | T | . | G | C | T | . | G | . | G | . | C | C | . | A | C | . | . | T | . |
| Traszka | HT27 | OR827129 | G | 3 | T | . | G | . | . | T | . | . | . | . | T | G | . | A | . | A | C | . | . | . | . | . | . | . | . | . | . | . | . | . | . | . | . | . | G | . | . | . | . | . | . | A | C | . | . | T | . |
| Traszka | HT28 | OR827130 | A | 2 | . | . | . | . | . | T | . | . | . | C | . | . | C | . | . | . | . | . | . | . | . | . | . | . | . | . | T | . | . | . | . | . | T | . | . | . | . | . | . | . | T | A | C | . | . | T | . |
| Traszka | HT29 | OR827131 | A | 1 | . | A | . | . | . | T | A | . | . | . | . | . | . | . | . | . | . | C | . | . | . | . | . | T | . | . | . | . | . | . | . | . | C | . | . | . | . | . | C | C | . | A | . | . | . | . | . |
| Tunguska | HT30 | OR827132 | A | 13 | . | . | . | . | . | . | . | . | . | . | . | . | . | . | . | . | . | . | . | . | . | . | . | . | . | . | . | . | . | . | . | . | . | . | G | . | . | . | . | . | . | . | . | . | . | . | . |
| Tygryska | HT31 | OR827133 | A/B | 5 | . | . | . | . | . | . | . | . | . | C | . | G | . | A | . | . | . | . | . | . | . | . | . | . | . | . | . | . | . | . | . | . | . | . | G | . | . | . | . | . | . | . | . | . | . | . | . |
| Tygryska | HT32 | OR827134 | A | 1 | . | A | . | . | . | T | A | . | . | . | . | . | . | . | . | . | C | A | T | . | T | . | T | . | . | . | T | . | T | . | G | . | T | . | G | . | G | . | . | . | . | A | . | . | . | . | . |
| Tygryska | HT33 | OR827135 | A | 1 | . | A | . | . | . | T | A | . | . | . | . | . | . | . | . | . | C | A | T | C | T | . | T | G | . | . | T | . | T | . | G | C | T | . | G | . | G | . | C | C | . | A | . | . | . | . | . |
| Urszulka | HT34 | OR827136 | A/B | 10 | T | . | G | . | . | T | . | . | . | . | . | G | . | A | . | A | C | . | . | . | . | . | . | . | . | . | . | . | . | . | . | . | . | . | G | . | . | . | . | . | . | . | . | . | . | . | . |
| Urszulka | HT35 | OR827137 | G | 9 | T | . | G | . | . | T | . | . | . | . | T | G | . | A | . | A | C | . | . | . | . | . | . | . | . | . | . | . | . | . | . | . | T | T | . | . | . | . | . | . | . | . | . | . | . | . | . |
| Urszulka | HT36 | OR827138 | A | 2 | . | . | G | . | . | T | . | . | . | . | . | G | . | A | . | A | C | . | . | . | . | . | . | . | . | . | . | . | . | . | . | . | T | . | . | . | . | . | . | . | . | . | . | . | . | . | . |
| Urszulka | HT37 | OR827139 | A/B | 1 | . | A | . | . | . | . | . | . | . | . | . | . | . | G | . | A | C | A | T | C | T | . | T | G | . | . | . | . | T | . | G | C | T | . | . | . | G | . | C | C | . | . | . | . | . | . | . |
| Wola | HT38 | OR827140 | A | 6 | . | . | G | . | . | T | . | . | . | . | . | . | . | . | . | . | C | . | . | . | . | . | . | . | . | . | . | . | . | . | . | . | . | . | G | . | . | . | . | . | . | A | . | . | . | C | . |
| Wola | HT39 | OR827141 | A | 5 | . | . | G | . | . | T | . | . | . | . | . | . | . | . | . | . | C | A | T | C | T | . | T | G | . | . | T | . | T | . | G | C | T | . | G | . | G | . | C | C | . | A | . | . | . | C | . |
| Zaza | HT20 | OR827122 | A | 7 | . | A | . | . | . | T | A | . | . | . | . | . | . | . | . | . | . | C | . | . | . | . | . | . | . | . | . | . | . | . | . | . | . | . | G | . | . | . | . | . | . | A | . | . | . | . | . |
| Zaza | HT33 | OR827135 | A | 5 | . | A | . | . | . | T | A | . | . | . | . | . | . | . | . | . | C | A | T | C | T | . | T | G | . | . | T | . | T | . | G | C | T | . | G | . | G | . | C | C | . | A | . | . | . | . | . |
| Zaza | HT40 | OR827142 | G | 4 | T | A | G | . | . | T | . | G | . | . | T | G | . | A | . | A | C | . | . | . | . | . | . | . | . | . | . | . | . | . | . | . | T | . | . | . | . | . | . | . | . | . | . | . | . | C | . |
| Zaza | HT41 | OR827143 | A | 3 | . | A | . | . | . | T | A | . | . | . | . | . | . | . | . | . | C | A | . | C | . | . | . | G | . | . | . | . | . | . | . | . | C | . | . | . | . | . | C | C | . | A | . | . | . | . | . |
| Zaza | HT42 | OR827144 | A | 1 | . | A | . | . | . | . | . | . | . | . | . | . | . | G | . | A | . | . | . | . | . | . | . | . | T | . | G | . | . | . | T | . | . | . | . | . | . | G | . | . | . | A | . | . | . | . | . |
| Zaza | HT43 | OR827145 | A | 1 | . | A | . | . | . | T | A | . | . | . | . | . | . | . | . | . | C | . | . | . | . | . | . | T | . | . | G | . | . | . | . | . | T | . | . | . | . | G | . | . | . | A | . | A | . | . | . |

**Note:**

*\* Achilli et al. (2012)*

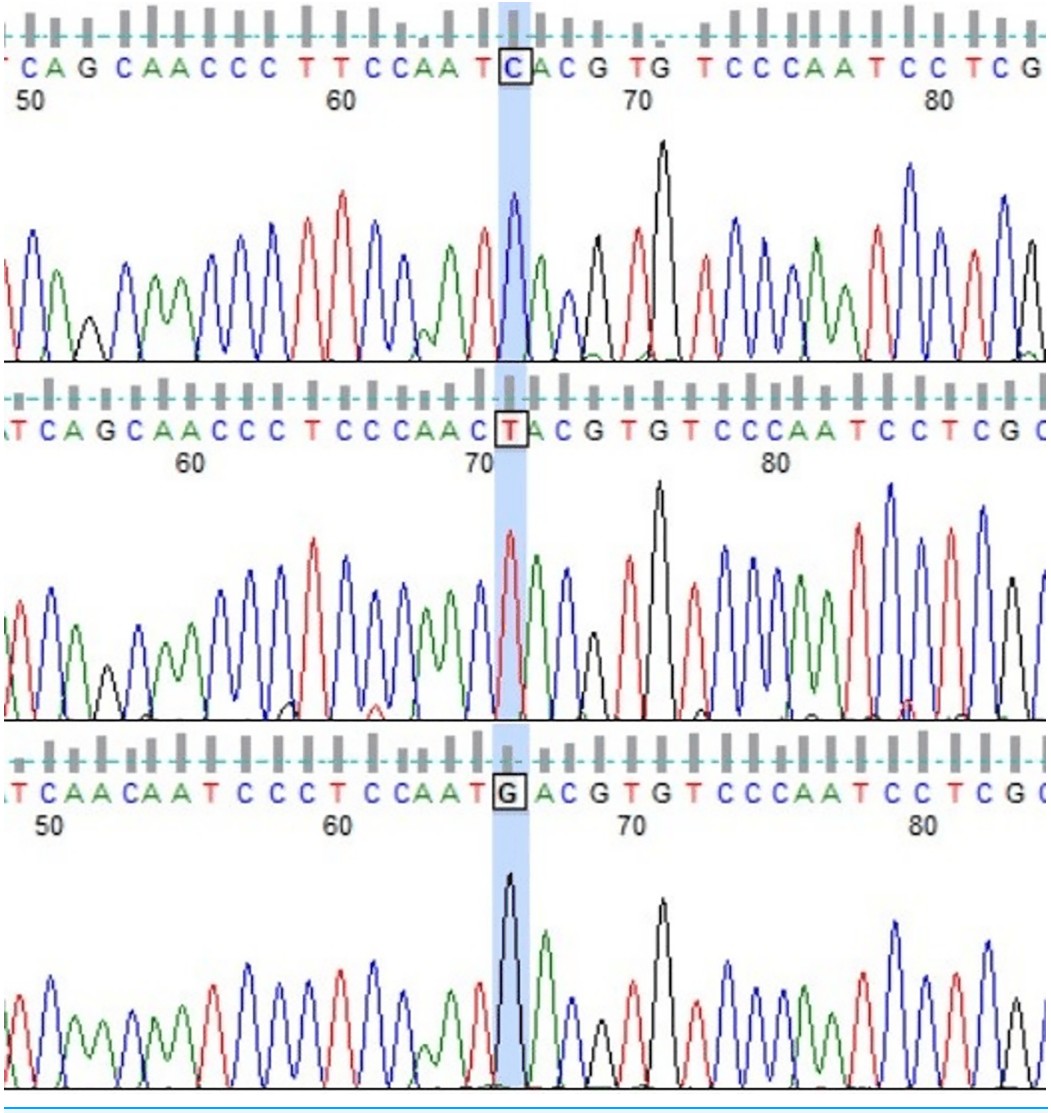

**Figure 2 Chromatograms showing three Polish Konik mtDNA HTs with marked 15,776 variable site position (FinchTV software).** The transition g.15776T>C was observed in HT3, the transversion g.15776T>G in HT7, and nucleotide T (in accordance with the reference sequence) in HT1.

dam lines and the *Equus caballus* (NC_001640) sequence as the root. The NJ tree presents the clustering of the analysed horses in separate groups with varied horse numbers (Table 2). Although most of the lines form clear clusters, some different sequences can be found in them; for example, the Tarpanka I_19 horse is embedded within the Popielica cluster. This finding suggests that the official Polish Konik pedigree data does not agree with the mtDNA HTs revealed from genetic analysis.

The median-joining network was constructed using all Polish Konik sequences to demonstrate the genetic distance between the obtained haplotypes (Fig. 4). The network indicated clusters' separation based on the number of polymorphic sites. Several dam lines

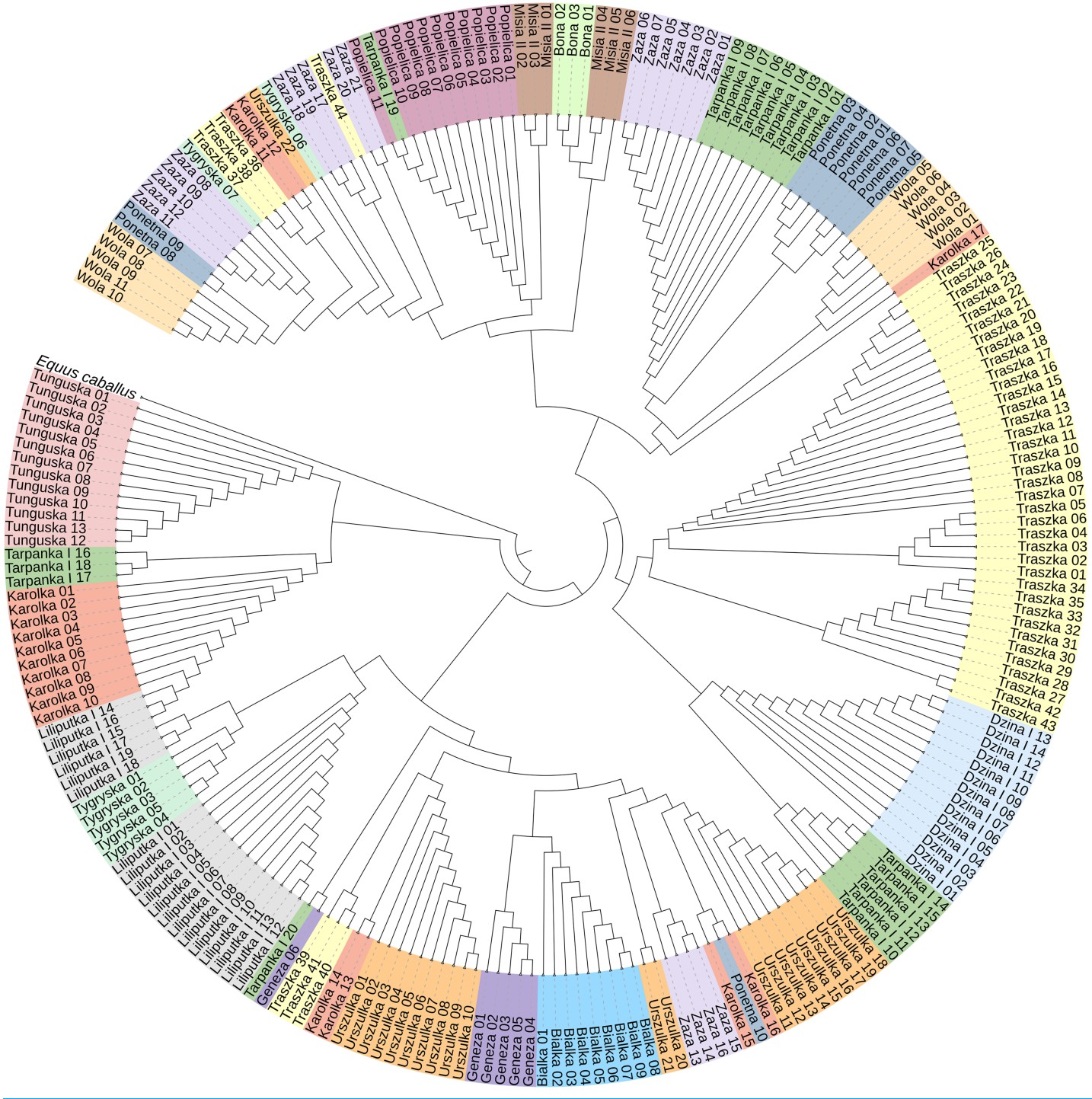

**Figure 3  The neighbor-joining tree of the 233 Polish Konik mtDNA sequences with *Equus caballus* reference sequence (NC_001640) used as a root.** Each colour represents one of sixteen dam lines.                               

formed HTs separated by a single polymorphism, which points out that their haplotypes are very similar. However, in some dam lines (including Karolka and Traszka lines), we observed several different, only distantly related HTs. The overview of both the NJ tree and

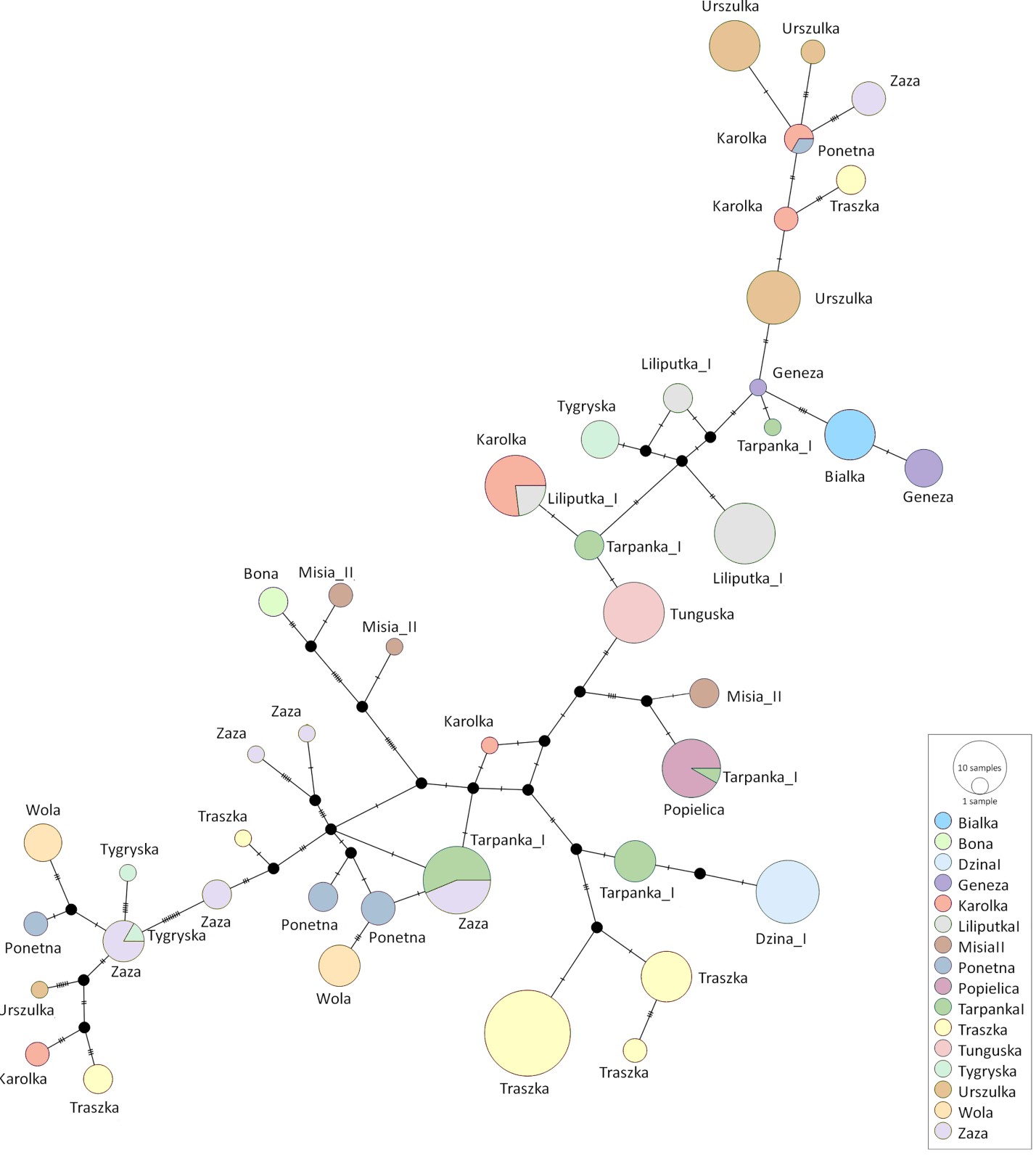

**Figure 4** **The median-joining network of the identified Polish Konik haplotypes (Table 2) with each dam line represented by circles coloured according to the dam line info from the pedigree.** The size of the nodes corresponds to the number of samples and the strokes on the branches correspond to the number of polymorphisms.

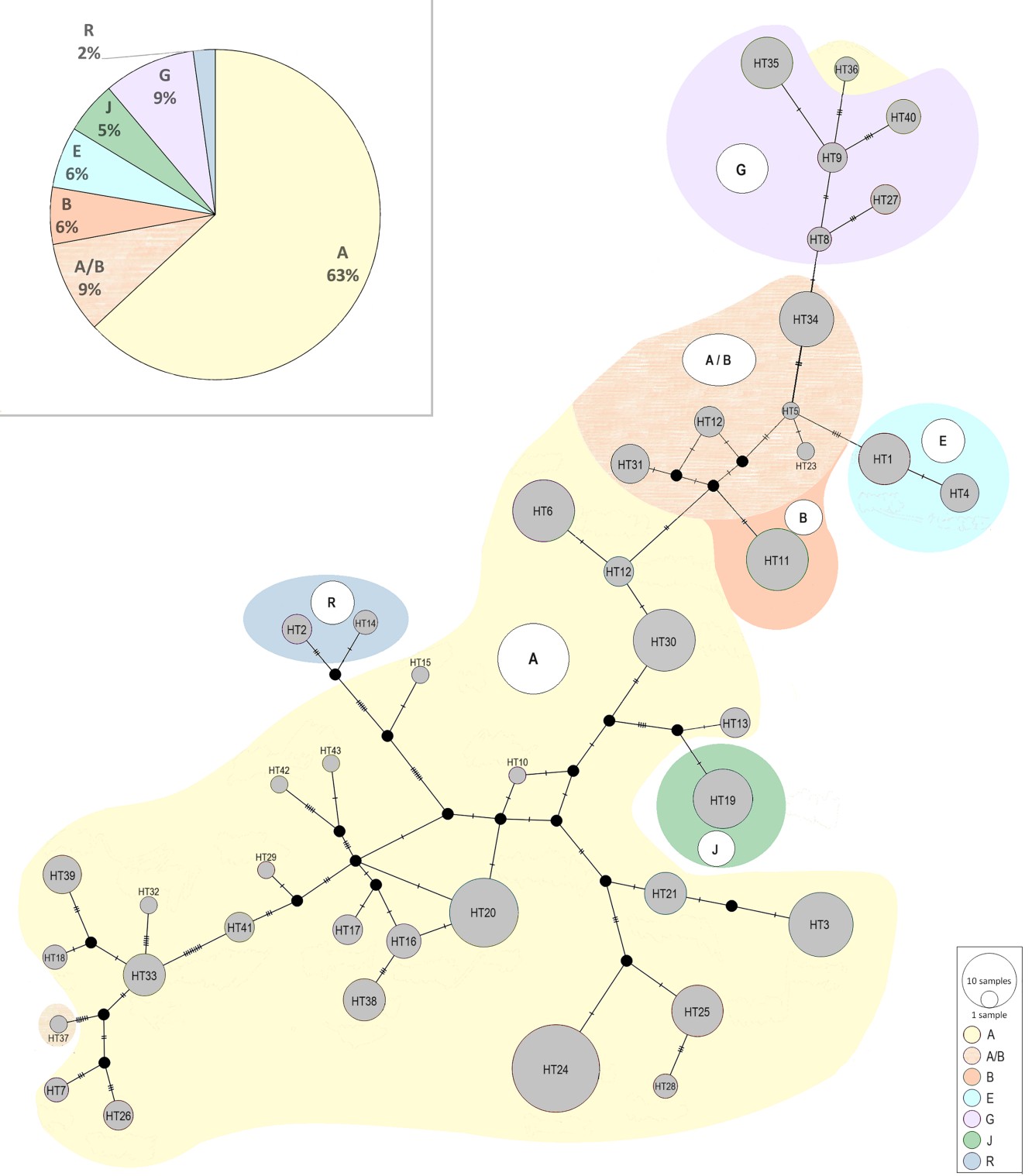

**Figure 5 The median-joining network of the identified Polish Konik haplotypes (Table 2) with their belonging to the generally known haplogroups described by** *Achilli et al. (2012)*. The size of the nodes corresponds to the number of samples. Six haplogroups are detected among the Polish Konik population: A, B, E, G, J, and R. The percentage share of detected haplogroups is presented in a pie chart.

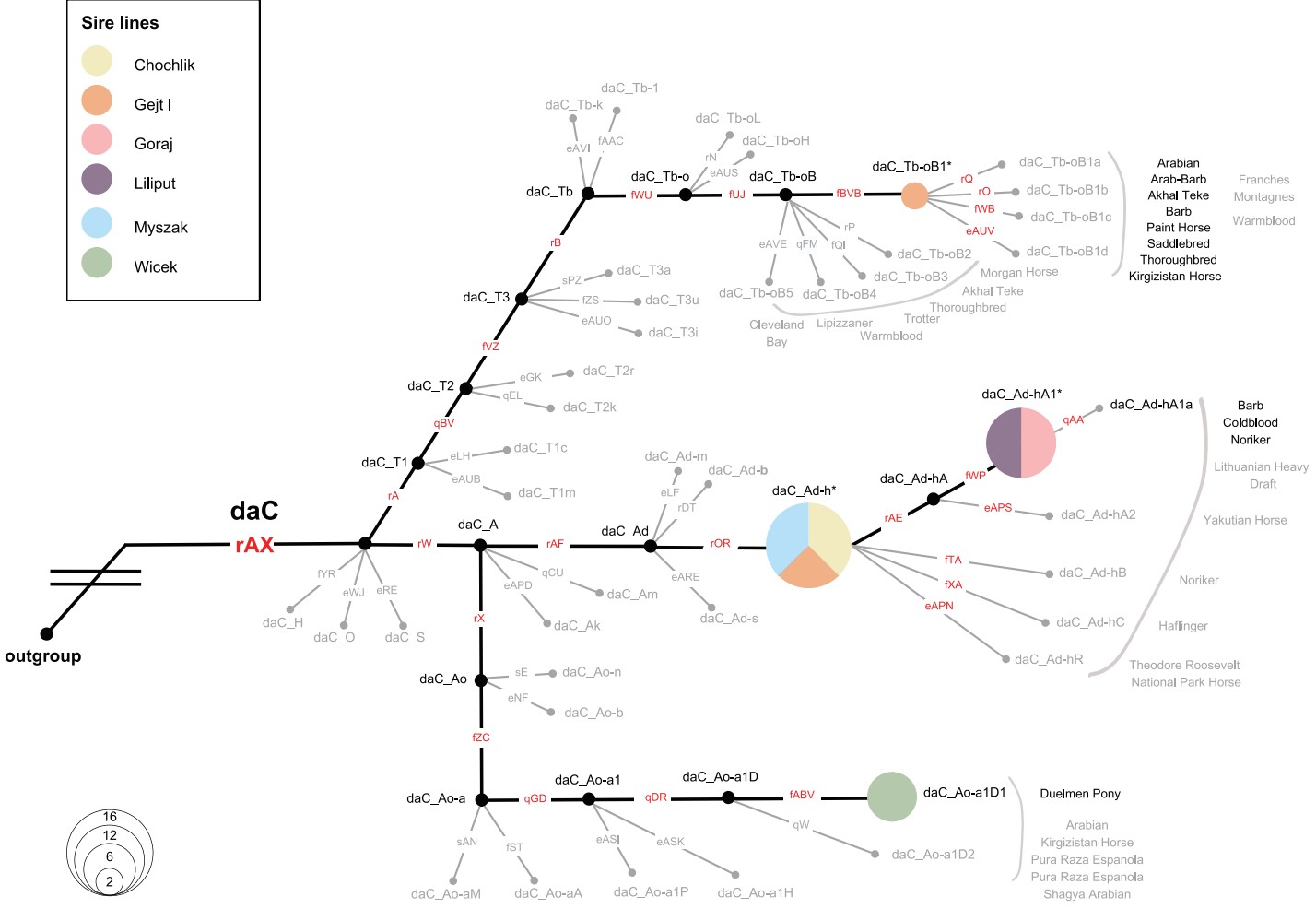

**Figure 6 Magnified view on the MSY HTs in Polish Konik sire lines.** The simplified haplotype structure based on 114 (92 Crown) variants (see variant details in Table S2), according to the published MSY topology (*Bozlak et al., 2023*). Variant names are placed on each branch, in red when tested in the sample set and in gray if their allelic states were imputed. Genotyping results from 36 males are shown as pies, sized according to frequency and colored with regard to the sire line. Undetected HTs are shown in gray and collapsed, and '*HTs' are placed on the respective branching points. Previously reported horse breeds that carry the same HTs as Polish Konik males are denoted in proximity in black, while gray-colored breeds indicate horse breeds reported in neighboring HTs (*Wallner et al., 2017*; *Felkel et al., 2019*; *Remer et al., 2022*; *Radovic et al., 2022*; *Bozlak et al., 2023*).

median-joining network highlights the wide range of variability among different mtDNA haplotypes in Polish Koniks.

The median-joining network above was also constructed in a way to presents all recognized haplotypes and their belonging to the generally known haplogroups described by *Achilli et al. (2012)*; (Fig. 5). Among the Polish Konik population were six haplogroups detected: A, B, E, G, J, and R. Similarly to the research conducted by *Cieślak et al. (2017)* haplogroup L, which is the most frequent in Europe and the Middle East, was not found. The haplogroup A observed as common among Asian horse populations (*Achilli et al., 2012*), turned out to be the most frequent within the population including 63–72% of all analysed horses. The presence of the other haplogroups ranged from several to a dozen percent. It should be noted that 9% of horses could not be clearly assigned to a specific

haplogroup because of sharing the same similarity to the haplogroup A and haplogroup B. It is the result of a smaller number of nucleotides used in our study than analysed in studies by *Achilli et al. (2012)*.

## Y chromosome variability

We investigated the MSY HTs in 36 Polish Konik males, representing all six sire lines. All samples were allocated to the Crown haplogroup (daC), where 4 HTs were distinguished. Notably, eight analysed males carried previously defined HTs (daC_Tb-oB1* and daC Ao-aA1D1), whereas 28 males were placed at internal nodes of the backbone topology (daC_Ad-h* and daC_Ad-hA1*; see Fig. 6 and Table S2), and their private HTs are not yet definitely defined in the genotyping backbone. The HT name of a sample that clustered internally was marked with an asterisk (*), *e.g.*, "daC_Ad-h*" (Fig. 6).

The most abundant HT was daC_Ad-h*, detected in 16 horses. It was noted, that all representatives of the Chochlik and Myszak sire lines clustered into daC_Ad-h*, while four individuals from the Glejt I stallion line carried this HT. A subsequent daC_Ad-hA1* HT was carried by twelve individuals, accounting for all analysed males of Liliput and Goraj sire lines. Interestingly, daC_Ad-h* and daC_Ad-hA1* HTs were most frequently reported in Coldbloods (*Felkel et al., 2019*; *Remer et al., 2022*; *Bozlak et al., 2023*). The remaining part of the dataset (22%) carried daC_Ao-a1D1 (six samples from the Wicek sire line) and daC_Tb-oB1* (two samples from the Gejt I sire line) HTs, respectively. Previously, HT daC_Ao-a1D1 was reported only in Duelmener Ponies (*Remer et al., 2022*). In contrast, daC_Tb-oB1* HT was characterized in Thoroughbreds (*via* Byerley Turk, 1680. founder), Arabians, Akhal Teke, Barbs, and many other horse breeds (*Wallner et al., 2017*; *Felkel et al., 2019*; *Remer et al., 2022*; *Radovic et al., 2022*; *Bozlak et al., 2023*).

## DISCUSSION

The analysis of population structure in unique, native horse breeds such as Polish Konik is extremely important because it allows the attempt to increase their genetic diversity and keeps all lines large enough to prevent them from extinction. Currently, in a situation where more than half of the Polish Konik lines are already extinct and several subsequent lines are threatened with extinction, an especially important task is to control the population structure and stop the regression of the least numerous lines in the population. Tools used for this purpose are mitochondrial DNA and Y chromosome sequence analysis.

The variability of mtDNA in modern horse breeds is generally high. For the first time, the mtDNA variation within the Polish Konik maternal lines was examined in 2017 (*Cieślak et al., 2017*). The research was conducted on 173 horses and included 510 bp, containing hypervariable region 1. The 33 variable sites were used to segregate Polish Koniks in the form of 19 haplotypes. In our study, the number of polymorphic sites used was 14 sites greater (47 SNPs), and this resulted in more than twice the number of detected haplotypes (43). This comparison may indicate that applying the entire hypervariable region gives us more detailed results than using only hypervariable region 1. Moreover, *Cieślak et al. (2017)*, for the first time, suggested that mtDNA results do not fully correspond to the official pedigree data. Another study focused on assessing the genetic

diversity and structure of the Polish Konik breed based on 17 microsatellite markers (*Fornal et al., 2021*), and this presented no signs of inbreeding and the assignment of the horses using Structure software into (most likely) three clusters. However, the study conducted by *Mackowski et al. (2015)* found 8.6% of inbreeding. The above studies showed the diversity of the Polish Konik population in terms of STR variability.

The occurrence of a small number of differentiating sites between HTs observed within a dam line may be the result of spontaneous mutations naturally occurring in the sequence over the years. On the other hand, the presence of clearly different haplotypes in one dam line can be the consequence of migration between different Polish Konik populations before the establishment of the studbooks or the presence of errors in the pedigrees. Further, both the neighbor-joining tree and the median-joining network demonstrated moderate variability among different mitochondrial DNA sequences and overall diverse variability of mitochondrial DNA sequence in Polish Konik dam lines.

The median-joining network constructed in the way to present the Polish Konik haplotypes belonging to the generally known haplogroups (*Achilli et al., 2012*) revealed the presence of six haplogroups (A, B, E, G, J, and R) among the Polish Koniks, which indicates multiple origins of dam lines. The most numerous haplogroup A including 63–72% of analysed horses was observed as most common among Asian horse populations. Taking into consideration the geographical attribution of the haplogroups, the A, E, G, and J haplogroups were the ones the most common in Asia, including 83–92% of tested Polish Koniks. The remaining two haplogroups B and R were most common for the North American and European populations (*Achilli et al., 2012*), respectively, and included 6–15%, and 2% of horses. These results, with an outstanding percentage predominance of haplogroups most common in Asia, may shed light on the unexplored importance of Asian horses in the development of the Polish Konik breed.

Another interesting observation noticed also by *Cieślak et al. (2017)* is that almost all of the dam lines represented by more than one haplotype belonged to (at least) two distinct haplogroups, which strengthens the assumptions regarding the possible presence of errors in the Polish Konik pedigrees.

In the case of paternal inheritance, 17 autosomal microsatellite loci were previously investigated among Polish Konik sire lines (*Fornal et al., 2020*). The microsatellite analysis showed low genetic diversity and vague population structure among sire lines, where the Polish Konik population can be described with only two clusters. The diversity of Y chromosome in modern horses is also very low, however, recent advances in horse MSY phylogeny (*Felkel et al., 2019*; *Bozlak et al., 2023*) enabled investigation and fine-scaled insights into the horse patrilines. Numerous studies used the stable structure of MSY phylogeny and investigated genetic variation (*e.g.*, *Wallner et al., 2017*), population history (*e.g.*, *Remer et al., 2022*), and breeding practices development (*e.g.*, *Castaneda et al., 2019*) of diverse horse breeds. Thus, the investigation of Y chromosomal markers is crucial to understanding the history and genetic patterns of Polish Konik patrilines.

Here, the MSY HT spectrum in Polish Koniks was investigated, and all sire lines were reported within the Crown group, which reflects the very recent breeding history (last 1,500 years) of the population (*Bozlak et al., 2023*). Among all analysed stallions, four

different haplotypes were noted, which exhibit a greater resolution of patriline differentiation than observed with microsatellite loci (*Fornal et al., 2020*). Interestingly, members of five out of six sire lines carried haplotypes across the daC_A clade (daC_Ao-a1D, daC_Ad-h*, and daC_Ad-hA1*). Among those, all Wicek line stallions carried daC_Ao-a1D1, previously described in Duelmener horses. This finding sheds new light on the close kinship between the breeds. Right after the end of World War II, Polish Konik stallions were used to eliminate Duelmener ponies' domesticated traits. A notable example was the stallion Nugat XII, who covered Duelmener mares in the years 1957–1963 (*Opora, 2006*). The haplotypes of Chochlik, Myszak, and Glejt I line in daC_Ad-h*, as well as the joint allocation of Goraj and Liliput males to daC_Ad-hA1*, indicated private, not yet resolved HTs in these lines. Previous research traced these HTs in Coldbloods (*e.g.*, *Felkel et al., 2019*; *Remer et al., 2022*), which is in line with the Polish Koniks' breeding history, phenotypic characteristics, and horse type itself since the breed is considered a 'light draft breed' (*Hendricks, 2007*). It is also noteworthy that a few males in the Glejt I lineage carried the daC_Tb-oB1* haplotype (Fig. 6), previously described in various lighter breeds. daC_Tb-oB1* was primarily attributed to the Thoroughbred founder Byerley Turk (1680) (*Felkel et al., 2019*) but later also detected in Arabians', Kuhailan Afas', and Latif' sires (*Remer et al., 2022*), as well as three patrilines in North African horses (*Radovic et al., 2022*). The MSY patterns and narrative on the vast use of Thoroughbred and Arabian stallions for the improvement of local stocks (*Hendricks, 2007*) could indicate undocumented crossbreeding, probably to expand the degree of genetic variability of the Polish Konik breed.

The described mitochondrial DNA and the Y chromosome genetic diversity also indicated a certain probability of pedigree errors in the Polish Konik due to the unique parental inheritance mechanism of mtDNA and the Y chromosome. Mitochondrial DNA analysis allowed us to highlight individuals whose haplotype fits perfectly into haplotypes occurring more frequently in a different dam line. In addition, MSY analysis emphasized the accuracy and information content of sire line tracking in the Wicek line. In contrast, we also observed two very distinct HTs in the Glejt I patriline (two males in daC_Tb-oB1* and four in daC_Ad-h*). This finding cannot be explained by *de novo* mutations but rather shows a clear sign of two different sire lines among analysed individuals from this line.

## CONCLUSIONS

Genetic analysis of Polish Konik dam and sire lines has revealed their diversity in the form of the identified haplotype number. The identification of mtDNA haplotypes among individual dam lines showed that in some of them only one single haplotype was presented, but several showed a wide range of haplotypes within the line. The haplotypes were also classified into six (A, B, E, J, G, R) haplogroups, with an outstanding percentage predominance of haplogroups most common in Asian horse populations. Y chromosome analysis revealed that most of the Polish Konik sire lines share their haplotype with Coldblood breeds, but also uncovered intriguing historical relationships and potential crossbreeding events. The possibility of errors in the Polish Konik pedigrees was also

reported. The findings emphasize the importance of genetic monitoring to maintain and control the genetic diversity of Polish Koniks and ensure their long-term survival.

### Funding
This research was funded by 'Diamentowy Grant' no. 0211/DIA/2019/48–Ministry of Science and Higher Education, Poland. The funders had no role in study design, data collection and analysis, decision to publish, or preparation of the manuscript.

### Grant Disclosures
The following grant information was disclosed by the authors:
Diamentowy: 0211/DIA/2019/48.
Ministry of Science and Higher Education, Poland.

### Competing Interests
The authors declare that they have no competing interests.

### Author Contributions
- Adrianna Dominika Musiał conceived and designed the experiments, performed the experiments, analyzed the data, prepared figures and/or tables, authored or reviewed drafts of the article, funding acquisition, and approved the final draft.
- Lara Radović conceived and designed the experiments, performed the experiments, analyzed the data, prepared figures and/or tables, authored or reviewed drafts of the article, and approved the final draft.
- Monika Stefaniuk-Szmukier conceived and designed the experiments, authored or reviewed drafts of the article, and approved the final draft.
- Agnieszka Bieniek conceived and designed the experiments, authored or reviewed drafts of the article, material collection, and approved the final draft.
- Barbara Wallner conceived and designed the experiments, analyzed the data, authored or reviewed drafts of the article, and approved the final draft.
- Katarzyna Ropka-Molik conceived and designed the experiments, performed the experiments, analyzed the data, authored or reviewed drafts of the article, and approved the final draft.

### Data Availability
All obtained mitochondrial DNA haplotypes are available at GenBank: OR827103–OR827145.

### Supplemental Information
Supplemental information for this article can be found online at http://dx.doi.org/10.7717/peerj.17549#supplemental-information.

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
