# Peer review of "Mitochondrial DNA and Y chromosome reveal the genetic structure of the native Polish Konik horse population"

_PeerJ, doi:10.7717/peerj.17549_

## Round 0.1 · original submission · Major Revisions

The manuscript by Musiał et al. is interesting, however, the authors should include
detailed analysis to classify samples into mtDNA haplogroups in the revised manuscript.

Reviewer 1 ·

Basic reporting

The authors of study entitled “Mitochondrial DNA and Y chromosome reveal the genetic structure of the Polish Konik horse breed” aimed to evaluate the genetic structure of Polish Konik Horse breed, by analyzing the mtDNA and MSY variation in a total of 233 samples. The paper is well written and easily readable, but there are some points that must be deepened, explained and modified.
Line 92-100: please, add at least a reference for these statements
Line 109-110: Authors should add some relevant references as examples
Line 113-115: there are many other references that should be cited here

Experimental design

In general, the weak link of this manuscript is the lack of a classification in mtDNA haplogroups and a comparative analysis with other horse breeds, as instead done for the male counterpart.

Validity of the findings

1. A detailed phylogeny was constructed for the horse maternal lines and I believe that authors should perform this analysis and classify all samples into mtDNA haplogroups.
Then, I suggest that authors perform a comparative analysis by including published data from other breeds and considering at least their haplogroup frequencies. For the sire lines, a comparative analysis with other horse breeds was performed.
2. Line 208: this part lacks of a statistical evaluation of the genetic diversity within the breed (i.e. haplotype diversity, nucleotide diversity, average number of nucleotide differences, etc). Please, perform also this analysis.
3. Line 212-213: Authors should compare at least these sequences to published data (i.e. by using BLAST) to understand if these dam lines are similar or identical to the HTs carried by other horse breeds.

Additional comments

Hereafter my minor comments:
Line 44: replace “Their body” with “Its body”
Line 48: replace “due to their” with “due to its”
Line 50: please, specify where they live in semi-feral and/or free-roaming conditions
Line 57: “about the origin” could be eliminated
Line 109: replace “see” with “detect”
Line 130: Authors talk about the entire control-region sequence, but in M&M section they state that the hypervariable regions 1 and 2 were analyzed. The horse D-loop is from 15469 to 16660 nps, where HVS-I is from 15,469 to 15,834 nps and HVS-II is in the range 16,351-16,660. Please, explain and modify also in the abstract.
Line 170: in general, for horse mtDNA, the Reference Sequence NC001640 is mostly used
Line 325-327: This statement is quite obvious, since there is not a direct correspondance between mtDNA HT and dam line, and vice versa. For this reason it could be very useful to evaluate a mtDNA haplogroup classification.

Reviewer 2 ·

Basic reporting

• Introduction, lines 72-73: change to “…..origin of the breed”
• Discussion, line 308: should be “Polish Konik stallions were USED to eliminate…”
• Discussion, line 309: please change as “A notable example was stallion Nugat XII…”.
• Discussion, line 313: please replace “concordance” with “in line with”.
• Discussion, line 315: should be “It is also noteworthy that a few males in Glejt I lineage carried…”.
• Table 1: please indicate 5’-3’ direction of the primers.
• Please check the text for punctuation, particularly for missing commas.

Experimental design

all good

Validity of the findings

Valid

Additional comments

SUMMARY
The authors have done nice work on investigating the origins of maternal and paternal lineages of the Polish Konik horse breed by determining mtDNA and MSY haplotypes, respectively. Their findings are in line with recent ancient DNA studies refuting the theory that Polish Konik is a descendant of the extinct Tarpan. Instead, the authors connect the existing lineages of Polish Konik to known domestic horse breeds/populations.
COMMENTS
1. The authors write about ‘unique genetic diversity” (Discussion line 264) of Polish Konik breed. Could you please support this statement with more evidence.
2. In Conclusions (line 334) the authors write “the degree of genetic diversity” but do not expand it, i.e., is it high, low, or medium compared to other horse breeds?
MINOR COMMENTS
• Introduction, lines 72-73: change to “…..origin of the breed”
• Discussion, line 308: should be “Polish Konik stallions were USED to eliminate…”
• Discussion, line 309: please change as “A notable example was stallion Nugat XII…”.
• Discussion, line 313: please replace “concordance” with “in line with”.
• Discussion, line 315: should be “It is also noteworthy that a few males in Glejt I lineage carried…”.
• Table 1: please indicate 5’-3’ direction of the primers.
• Please check the text for punctuation, particularly for missing commas.

Reviewer 3 ·

Basic reporting

The manuscript focuses on examining the mitochondrial DNA D-loop sequence and MSY variability looking at dam and sire founding lines in Polish Koniks of Poland. It provides important information on the genetic makeup of 233 Konik mares and 36 stallions.

The manuscript is generally well-written, professional English language use is good, with only a few ambiguities.
The introduction gives a nice overview of the topic, and likely makes the background understandable for readers that are not familiar with the multifaceted problem-package around the Konik breed. However, the article itself clearly is more interesting for a relatively narrow target audience. (This is not a weakness though!)

I find good that the introduction gives a summary of the contested history of the origins of the Konik breed with references to the Tarpan, and it tries to be objective in this regard. After changing some overstatements, which I indicated in comments in the pdf, this part will facilitate future research referring to the origins of this horse breed.
I also appreciate some other details such as the precise numbers of the current Konik population registered in the Polish studbook. It would be an interesting addition how many of the individuals currently in the studbook are from Poland and what is the number of horses that are located abroad. Further, I think targeted readers who are conducting research on Koniks would be curious of the estimated number of Koniks (or at least the number of distinct populations of Koniks) that are not included in the studbook (both in Poland and abroad). Please try to give these, if possible.

I found some typos and a missing reference, please have one more look on the reference list.
I would also suggest to include a wider range of international literature (especially in the introduction).

Raw data is provided, it is clean and well-structured.

Figures are relevant and made with care to ease the reader’s understanding. I would however suggest to use the same color-code for the dam lines in Fig. 3 and 4.

Experimental design

The research is within the scope of PeerJ.

The research aims are clear, and the results respond to the aims.
The description of the analysis is detailed and well-structured. The applied methodology is clear from the description and overall, the methods appear thorough and rigorous.

There may be a few aspects where limitations of the applied methods may arise.
For example, because the assignment of horses to specific founding lines relies on pedigree data, inaccuracies or gaps in pedigree information could reduce the accuracy of the genetic analysis. I would suggest to discuss this limitation in the discussion.
Then, DNA extraction from hair vs blood samples may introduce biases in the results, because the quality of the DNA originating from hair or blood is not the same, which might affect the accuracy and reproducibility of results.
Also, was there a specific reason why you used Sanger sequencing?

Validity of the findings

Similar earlier research has been conducted about the genetic makeup of Koniks, but every new approach adds to the knowledge basis – this article is a also addition and future research will benefit from it. I would suggest to discuss the possible extension to Konik populations outside of Poland.

A major lacking element of the interpretation of the results is how the mtDNA and Y chromosome marker distribution in modern horse breeds may affect the validity of the findings. I commented about this in the pdf.

The data is provided as suppl. mat. and haplotypes deposited in GenBank, thus it could be of help of future research.
The data is robust enough to draw conclusions about the Polish population of the Koniks – but conclusions should not be generalized to the entire Konik breed. The sample was drawn only from the Polish population (more precisely, only from individuals that are included in the studbook), so I would restrict generalizing to this population. (Please also see my respective comments in the pdf.) Please also revise the title accordingly.

Annotated reviews are not available for download in order to protect the identity of reviewers who chose to remain anonymous.

---

## Round 0.2 · Minor Revisions

Authors are requested to follow strictly the concerns raised by the reviewer.

Reviewer 1 ·

Basic reporting

Authors of study entitled “Mitochondrial DNA and Y chromosome reveal the genetic structure of the Polish Konik horse breed” revised the manuscript accordingly to the majority of suggestions, but they did not perform a comparative analysis with other horse breeds. They simply described the worldwide distributions and frequency of mtDNA lineages.
The paper is well written and easily readable and some relevant references have been added.
Furthermore (point 6 of the Response file) authors declared “The same response as in the point 4” but they did not compare (at least!) their sequences to published data to understand if these dam lines are similar or identical to the HTs carried by other horse breeds. In M&M section, authors stated that “Haplotypes found within the studied Polish Konik population were compared with sequences stored in the GenBank database using BLAST (Altschul et al., 1990). The sequences representing haplogroups described by Achilli et al. (2012) were found among the BLAST results. The sequence with the highest similarity percentage was selected, and its haplogroup was checked to categorize the identified Polish Konik haplotypes into the haplogroups.” but they did not describe the results concerning the haplotypes, they talked only about the HG classification. It could be useful to evaluate the genetic peculiarities of the analysed population.

Experimental design

No more suggestions.

Validity of the findings

Point 6: Authors declared “The same response as in the point 4” but they did not compare (at least!) their sequences to published data to understand if these dam lines are similar or identical to the HTs carried by other horse breeds. It could be useful to evaluate the genetic peculiarities of the analysed population.

Additional comments

Hereafter my minor comments:
Line 20: replace “Polish Koniks remain” with “Polish Konik remains”
Line 24: variables?
Line 145: did authors mean “proportional to the total number of horses belonging to each dam line”?
Line 180: delete “sequences with detected”
Line 262-264: re-write the sentence “It should be noted that 9% of horses showed the same similarity with haplogroup A as with haplogroup B, which may result from a smaller sequence area selected for analysis than in Achilli et al. (2012) studies.” It is not clear

Reviewer 3 ·

Basic reporting

The manuscript improved after revision.

Experimental design

no comment

Validity of the findings

no comment

Additional comments

I suggest a small change in the title: Mitochondrial DNA and Y chromosome reveal the genetic structure of a Polish Konik horse population.
(I.e., there are several Konik populations and the authors looked at one population.)

---

## Round 0.3 · accepted · Accept

The authors have completely revised the manuscript in light of suggestions made by the reviewers. I feel happy to accept this manuscript for publication.